# Sparsomycin Exhibits Potent Antiplasmodial Activity In Vitro and In Vivo

**DOI:** 10.3390/pharmaceutics14030544

**Published:** 2022-02-28

**Authors:** Nanang Rudianto Ariefta, Baldorj Pagmadulam, Coh-ichi Nihei, Yoshifumi Nishikawa

**Affiliations:** 1National Research Center for Protozoan Diseases, Obihiro University of Agriculture and Veterinary Medicine, Inada-cho, Obihiro 080-8555, Japan; nanang.ariefta@gmail.com (N.R.A.); pagmadulam1028@gmail.com (B.P.); 2Laboratory of Microbial Synthesis, Institute of General and Experimental Biology, Mongolian Academy of Sciences, Ulaanbaatar Peace Avenue-54b, Ulaanbaatar 13330, Mongolia; 3Institute of Microbial Chemistry (BIKAKEN), 3-14-23 Kamiosaki, Shinagawa-ku, Tokyo 141-0021, Japan; conihei@bikaken.or.jp

**Keywords:** sparsomycin, antiplasmodial, *Plasmodium falciparum*, *Plasmodium yoelii*, *Plasmodium berghei*

## Abstract

The emerging spread of drug-resistant malaria parasites highlights the need for new antimalarial agents. This study evaluated the growth-inhibitory effects of sparsomycin (Sm), a peptidyl transferase inhibitor, against *Plasmodium falciparum* 3D7 (chloroquine-sensitive strain), *P. falciparum* K1 (resistant to multiple drugs, including chloroquine), *P. yoelii* 17XNL (cause of uncomplicated rodent malaria) and *P. berghei* ANKA (cause of complicated rodent malaria). Using a fluorescence-based assay, we found that Sm exhibited half-maximal inhibitory concentrations (IC_50_) of 12.07 and 25.43 nM against *P. falciparum* 3D7 and K1, respectively. In vitro treatment of *P. falciparum* 3D7 with Sm at 10 or 50 nM induced morphological alteration, blocked parasites in the ring state and prevented erythrocyte reinvasion, even after removal of the compound. In mice infected with *P. yoelii* 17XNL, the administration of 100 μg/kg Sm for 7 days did not affect parasitemia. Meanwhile, treatment with 300 μg/kg Sm resulted in a significantly lower parasitemia peak (18.85%) than that observed in the control group (40.13%). In mice infected with *P. berghei* ANKA, both four and seven doses of Sm (300 μg/kg) prolonged survival by 33.33%. Our results indicate that Sm has potential antiplasmodial activities in vitro and in vivo, warranting its further development as an alternative treatment for malaria.

## 1. Introduction

Malaria is a mosquito-borne infectious disease caused by protozoan parasites of the genus *Plasmodium*. Approximately half of the global population is at risk for the disease, especially socio-economically disadvantaged populations [1]. Based on the World Malaria Report 2021, an estimated 241 million malaria cases occurred globally in 2020, up from 227 million in 2019, with most of this increase arising from countries in the African region. Conversely, malaria deaths increased by 12% versus 2019 to an estimated 627,000. The increase in 2020 was associated with the disruption of malaria prevention, diagnosis and treatment during the COVID-19 pandemic [2].

One of the threats to efforts to eradicate malaria is the emergence of drug-resistant parasites [3]. For example, chloroquine-resistant *P. falciparum* and *P. vivax* spread from 1957 to 1987 through Asia, South America and Africa and, more recently, *P. falciparum* resistant to artemisinin derivatives was identified in Southeast Asia, leading to significant rates of treatment failure for these gold-standard combination therapies [4]. The spread of multidrug-resistant malaria parasites is broadly caused by four factors, i.e., mutation and/or amplification of the target(s), stress response-based survival mechanisms and removal or sequestration of drugs and detoxifications [5]. Thus, the continuous development of novel antimalarial compounds is needed. Accordingly, we screened a compound library consisting of natural products and natural-product derivatives, curated by the Institute of Microbial Chemistry. From this screening, sparsomycin was stood out to exhibit antiplasmodial activity at the nanomolar scale and was further evaluated in this study.

Sparsomycin (Sm) is an antibiotic that was initially isolated from *Streptomyces sparsogenes* in 1962 [6]. Its structure contains a rare monoxodithioacetal moiety (Figure 1). Sm acts as a protein biosynthesis inhibitor in bacteria, archaea and eukaryotes and it selectively acts on several human tumors [7]. Sm interferes with peptide bond formation, a process that is essential to protein biosynthesis, on the large ribosomal subunit (50S and 60S) [8]. Sm binds primarily to the peptidyl (P) site and, because of the stabilization of the P-site/tRNA, the drug blocks the critical movement of tRNAs between the aminoacyl site and P-site, resulting in the inhibition of peptide bond formation and the inability of aminoacyl–tRNA to access the P-site [9,10,11]. Despite acute toxicity of Sm in mice [6] and Sm-related retinopathy in the phase I clinical study [12], an active testing program of the drug related to the protein biosynthesis machinery has continued [10,13,14]. Considering the effects of Sm on protein translation, the in vitro inhibitory activity of Sm against the most common and deadliest human malaria parasite, *P. falciparum* (3D7 and K1) [15], was evaluated. *P. falciparum* 3D7 represents a chloroquine-sensitive strain and *P. falciparum* K1 represents a multi-drug resistant strain, including chloroquine, sulphadoxine, pyrimethamine and cycloguanil [16]. Furthermore, to confirm the in vitro results, the effects of Sm were also evaluated in vivo using two rodent malaria parasites with different virulence characteristics, *P. yoelii* 17XNL and *P. berghei* ANKA, respectively, for uncomplicated/non-lethal and complicated/lethal strains [17].

## 2. Materials and Methods

### 2.1. Compounds

Sm (molecular weight (MW), 361.43 g/mol; CAS 1404-64-4, Cat. No. 29934) was purchased from Cayman Chemical (Ann Arbor, MI, USA). Chloroquine diphosphate (MW, 515.86 g/mol; CAS 50-63-5, Cat. No. C6628) and artemisinin (MW, 282.33 g/mol; CAS 63968-64-9, Cat. No. 361593) were purchased from Sigma-Aldrich (St. Louis, MO, USA). Stock solutions were prepared in MilliQ water for Sm (2.8 mM) and chloroquine diphosphate (20 mM) and in 100% DMSO for artemisinin (20 mM). The stocks were stored at −30 °C until use and diluted as required.

### 2.2. Parasites

This study used two strains of *P. falciparum* for in vitro analyses, namely, chloroquine-sensitive (3D7) and multidrug-resistant (K1) strains, obtained from Dr. Shin-ichiro Kawazu (National Research Center for Protozoan Diseases, Obihiro University of Agriculture and Veterinary Medicine, Japan). Two rodent malaria parasites were used for in vivo studies, namely, *P. yoelii* 17XNL (for uncomplicated rodent malaria) and *P. berghei* ANKA (for complicated rodent malaria), obtained from the Department of Molecular Parasitology, Ehime University Graduate School of Medicine, Japan.

### 2.3. In Vitro Culture of P. falciparum

*P. falciparum* strains 3D7 and K1 were cultivated in 2% washed human O-positive erythrocytes (supplied by Japanese Red Cross Society, Hokkaido, Japan) using a multi-gas incubator (37 °C, 5% CO_2_ and 5% O_2_). Roswell Park Memorial Institute (RPMI)-1640 medium (Cat. No. R6504, Sigma-Aldrich, St. Louis, MO, USA) containing 25 mM HEPES (CAS 7365-45-9, Cat. No. H4034; Sigma-Aldrich, St. Louis, MO, USA), 184 μM hypoxanthine (CAS 68-94-0, Cat. No. 086-03403; Wako, Osaka, Japan), 24 mM NaHCO_3_ (CAS 144-55-8, Cat. No. 198-01215; Wako, Osaka, Japan), 0.025% (*v*/*v*) gentamicin (50 mg/mL; CAS 1403-66-3, Cat. No. 15750078; Gibco, Waltham, MA, USA) and 0.5% (*w*/*v*) AlbuMax™ II Lipid-Rich BSA (Cat. No. 11021029; Gibco, Waltham, MA, USA), designated as complete medium, was used for the parasite culture. Parasitemia was monitored using Giemsa-stained thin blood smears (CAS 51811-82-6, Cat. No. 48900; Merck, Darmstadt, Hesse, Germany); the medium was changed daily and subculturing was performed as required.

### 2.4. In Vitro Inhibition Assay of P. falciparum

The in vitro growth inhibition of *P. falciparum* 3D7 and K1 was examined using the SYBR Green I-based fluorescence assay (SYBR^®^ Green I Nucleic Acid Stain 10,000×; Cat. No. 50513; Lonza, Basel, Switzerland) as previously described [18,19]. Briefly, test compounds were diluted in complete medium to eight desired concentrations (2-fold serial dilution). Next, the parasites were synchronized using treatment of 5% *D*-sorbitol (CAS 50-70-4, Cat. No. 191-14735; Wako, Osaka, Japan) to obtain ≥90% ring-stage parasites. Next, 50 μL of synchronous parasites at 0.5% parasitemia and 2% hematocrit was seeded in a 96-well plate containing 50 μL of test compounds. The 96-well plate was incubated for 72 h at 37 °C, 5% CO_2_ and 5% O_2_. Next, 100 μL of lysis buffer containing 0.02% (*v*/*v*) SYBR Green I was added to each well and mixed and then the 96-well plate was incubated at room temperature for 2 h in the dark. The fluorescence intensities were then measured using a Fluoroskan Ascent instrument (Thermo Fisher Scientific, Waltham, MA, USA) at excitation and emission wavelengths of 485 and 518 nm, respectively. Erythrocytes treated with 1% DMSO were used as a negative control and wells containing only test compounds and erythrocytes were used to correct background signals. In at least three independent trials, the inhibition assay was performed in quadruplicate for each concentration.

### 2.5. Microscopic Analysis of Parasitemia and Morphology of P. falciparum 3D7 Treated with Sm In Vitro

The parasites were tightly synchronized using a two-step sorbitol treatment protocol [20] and used at the ring stage with 0.5% parasitemia and 2% hematocrit. Sm (50 and 10 nM) in complete medium and 1% DMSO in complete medium were used as the treated and untreated groups, respectively, in this assay. Subsequently, the Giemsa-stained thin blood smears of each treatment group were prepared after 1, 24, 48, or 72 h of incubation. After 72 h, incubation was continued after changing the medium in either the presence (unwashed group) or absence (washed group) of Sm for up to 144 h. Subsequently, the Giemsa-stained thin blood smears of each treatment group were prepared at 96, 120 and 144 h. The blood smears were observed using a BZ-900 all-in-one microscope (Keyence BioRevo, Tokyo, Japan). The parasitemia levels were determined by enumerating the number of infected erythrocytes in relation to uninfected erythrocytes (a minimum of 500 cells were counted). The assay was performed in triplicate for each concentration and time point.

### 2.6. In Vitro Cytotoxicity in Human Cells

Cultures of human foreskin fibroblasts (HFFs) were maintained in Dulbecco’s modified Eagle medium (DMEM; Cat. No. D0819; Sigma, St. Louis, MO, USA) supplemented with 10% fetal bovine serum (FBS; Cat. No. S181A; Biowest, Riverside, MO, USA) and 1% penicillin–streptomycin solution (×100; Cat. No. 168-23191; Wako, Osaka, Japan) at 37 °C and 5% CO_2_. A cell viability assay was used to evaluate the cytotoxic action of Sm as described previously [18]. Briefly, a 100 μL cell suspension was seeded in a 96-well plate at a concentration of 1 × 10^5^ cells/mL in DMEM containing 10% FBS and incubated for 48 h at 37 °C and 5% CO_2_. Next, 2-fold dilutions (total of eight concentrations) of test compounds in DMEM were added in quadruplicate to each well and incubated for an additional 72 h. After that, a Cell Counting Kit-8 (CCK-8; Cat. No. 343-07623; Dojindo, Kumamoto, Japan) was added, cells were incubated for an additional 3 h at 37 °C and 5% CO_2_ and the absorbance was measured at 450 nm.

### 2.7. In Vitro Sm Hemolysis Rate in Human Erythrocytes

An erythrocyte hemolysis assay was performed as previously described [21]. Briefly, 100 μL of each compound (final concentration, 100 μM) in 1× phosphate-buffered saline (PBS) was seeded in a 96-well plate and 100 μL of a 3% erythrocyte suspension in PBS was added. The plate was incubated for 3 h at 37 °C, 5% CO_2_ and 5% O_2_ and then centrifuged at 1300× *g* for 5 min. Subsequently, 100 μL of the supernatant of each mixture was transferred to a new 96-well plate and the absorbance was recorded at 540 nm. PBS (containing 1% DMSO) and erythrocyte lysis buffer (0.83% NH_4_Cl; 0.01 M Tris-HCl, pH 7.2) served as the negative and positive controls, respectively. The erythrocyte hemolysis rate was calculated using the following formula: hemolysis rate = ([A_sample_ − A_negative control_]/[A_positive control_ − A_negative control_] × 100), where A is absorbance [21]. The experiments were performed in triplicate and repeated three times independently.

### 2.8. Mice and In Vivo Infections

Male C57BL/6 J mice (8 weeks old with body weight of 20–25 g; six mice per group) were purchased from Japan CLEA (Tokyo, Japan). The mice were allowed free access to water and food (CLEA Rodent Diet CE-2; Japan CLEA, Tokyo, Japan). The room temperature (24 °C), relative humidity (50%) and lighting (light from 8 a.m. to 8 p.m.) were adjusted and controlled. *P. yoelii* 17XNL and *P. berghei* ANKA were recovered from frozen packed erythrocyte stocks via passage in mice following intraperitoneal (i.p.) injection. Challenge experiments were performed with i.p. injections of 1 × 10^7^ fresh infected erythrocytes from donor mice (designated as 0 days post-infection (dpi)). Approximately 2 h after the injection, 2 μL of blood was collected from the tip of the mice’s tail and the parasitemia levels were checked using Giemsa-stained thin blood smears. When the parasitemia level reached 1%, the infected mice were intraperitoneally treated with vehicle (1 × PBS) or Sm at a dose of 100 or 300 μg/kg (modified from reported IC_50_ in Ref. [22]) once daily for 7 days (0–6 dpi; modified Peters’ test [23]) for *P. yoelii* 17XNL and 300 μg/kg once daily for 4 (0–3 dpi; Peters’ 4-day test [24]) or 7 days (0–6 dpi, [23]) for *P. berghei* ANKA. The survival and clinical signs of mice were monitored daily after infection. Mice with high clinical signs (arched back, immobility, >20% body weight loss and pain sign) were euthanized immediately. Parasitemia in *P. yoelii* 17XNL-infected mice was observed daily using Giemsa-stained thin blood smears (up to 30 dpi).

### 2.9. Statistical Analysis

Data were analyzed using GraphPad Prism 8 (GraphPad Software, Inc., La Jolla, CA, USA). IC_50_ or CC_50_ values were calculated from three independent experiments using a non-linear regression fit to logarithm concentration value of the compound versus inhibition percentage of parasite or cell growth, respectively. Group comparison analyses were performed using the one-way or two-way ANOVA followed by Tukey’s or Sidak’s multiple comparison test. Survival rates were calculated using the log-rank (Mantel–Cox) test. A *p*-value of <0.05 was considered statistically significant and is shown as an asterisk or a symbol, defined in each table/figure legend together with the name of the test used.

## 3. Results

### 3.1. Effects of Sm on P. falciparum In Vitro

The antiplasmodial activity of Sm in vitro versus artemisinin and chloroquine was determined using *P. falciparum* 3D7 and K1. As presented in Table 1, Sm inhibited the growth of both *P. falciparum* 3D7 and K1 with IC_50_ values of 12.07 ± 4.41 and 25.43 ± 8.15 nM, respectively. The cytotoxicity of Sm against HFFs was confirmed with a CC_50_ of 1.14 ± 0.03 μM, resulting in selectivity indices (SIs) of 94.45 (3D7) and 44.83 (K1). Furthermore, at a concentration of 100 μM, Sm induced an erythrocyte hemolysis rate of 1.04 ± 0.23%.

### 3.2. Effects of Sm on parasitemia and the Morphology of P. falciparum 3D7 In Vitro

A phenotype analysis of untreated and Sm-treated cultures of *P. falciparum* 3D7 was performed to investigate the parasitemia level and morphological changes under Sm exposure. Parasites were tightly synchronized using 5% *D*-sorbitol to the ring stage and incubated with 0, 10, or 50 nM Sm. Distinct differences were observed compared with the untreated group (Figure 2A). The life cycles of parasites were blocked in the ring stage over 1–72 h of incubation with 10 or 50 nM Sm. Sm treatment at 50 nM for 72 h induced morphological alterations, including shrinkage, as indicated by dot-like spots within erythrocytes. The parasitemia levels during this experiment were not increased along the incubation period (Figure 2B), indicating that reinvasion of the erythrocytes because of Sm underexposure did not occur. In cells cultured after 72 h of Sm exposure, the suppression of parasite growth continued in both the unwashed and washed groups and the parasites were continuously observed as shrunken dot-like spots within erythrocytes (Figure 3A). No increases in parasitemia levels were observed with prolonged incubation (Figure 3B).

### 3.3. Effect of Sm on P. yoelii 17XNL-Infected Mice

As presented in Figure 4A, 100 μg/kg Sm did not significantly alter parasitemia levels in *P. yoelii* 17XNL-infected mice versus the control. Meanwhile, 300 μg/kg Sm significantly reduced parasitemia compared with the control throughout the observation period (Figure 4B). Furthermore, a significantly lower parasitemia peak (18.85%) was observed in Sm-treated mice than in control mice (40.13%). No mortality was observed in either control or Sm-treated mice.

### 3.4. Effect of Sm on P. berghei ANKA-Infected Mice

On the basis of the effects of 300 μg/kg Sm on *P. yoelii* 17XNL-infected mice, the same dose was used in mice infected with *P. berghei* ANKA. Treatment for 4 or 7 days prolonged survival by 33.33% (up to 25 dpi). Although the effect of the 7-day regimen was statistically insignificant (*p* > 0.05), the effects of this regimen on survival were generally in accordance with the effects of the 4-day regimen.

## 4. Discussion

Sm is a well-known peptidyl transferase inhibitor that acts by stabilizing the P-site within large ribosomal subunits and it has been demonstrated to stimulate factor-independent translocation [25]. In the case of cancerous cells, which are more sensitive to translation inhibitors than normal cells because of the hyperactivation of the translation machinery [26], the high uptake of the compound resulted in inhibited protein biosynthesis, thereby causing metabolic disorders, inhibiting cell proliferation and promoting cell death [27]. In this study, Sm inhibited the in vitro growth of *P. falciparum* 3D7 and K1 (Table 1) after 72 h of exposure with IC_50_ values of 12.07 ± 4.41 and 25.43 ± 8.15 nM, respectively. The IC_50_ values were comparable to those of artemisinin (13.18 ± 2.66 nM for 3D7 and 19.89 ± 1.51 nM for K1) and better than those of chloroquine (26.20 ± 3.66 nM for 3D7 and 740.07 ± 95.67 nM for K1). Sm exhibited a lower resistance index (2.1) than chloroquine (28.24) against *P. falciparum* K1 [28]. Conversely, Sm exhibited cytotoxicity against HFFs with a CC_50_ of 1.14 ± 0.03 μM, resulting in SIs of 94.45 and 44.83 for 3D7 and K1, respectively. Sm meets the validated hit selection criteria for antimalarial compounds [29], including an IC_50_ of <1 μM for sensitive and resistant strains of *Plasmodium* spp., as well as greater than 10-fold selectivity between the CC_50_ for the mammalian cell line (HFFs in this study) and the IC_50_ for *Plasmodium* spp. Regarding cytotoxicity against human erythrocytes, Sm exhibited a low erythrocyte hemolysis rate at 100 μM (1.04 ± 0.23%), a concentration that was approximately 5000-fold higher than the IC_50_. This hemolysis rate falls with the safe range (<2%) recommended by the American Society for Clinical Pathology [30].

In vitro cultures of *P. falciparum* 3D7 exposed to 10 or 50 nM Sm for 1–72 h were blocked in the ring stage (Figure 2A), indicating that Sm might disrupt the *P. falciparum* ribosome and possibly its three translationally active subcellular compartments, i.e., cytosol, apicoplasts and mitochondria [31,32]. Similar retained phenotype profiles of *P. falciparum* 3D7 were observed following treatment with the protein translation inhibitors REP3123 and REP8839 [33]. Translation inhibitors, including azithromycin [34], doxycycline [35] and tetracyclines [36], have also exhibited great clinical success as potent antimalarials. The *Plasmodium* ribosomes do not fit into the prokaryotic–eukaryotic classification and they appear to be a mixed population of these ribosomes or evolutionary in the middle of prokaryotic–eukaryotic ribosomes (70S and 80S) [35]. Accordingly, a therapeutic window has resulted from the distinction between *Plasmodium* cytoplasmic ribosomes and the human ribosomes.

After 72 h of exposure to Sm (Figure 3A), cells were incubated for an additional 72 h in the presence (unwashed group) or absence (washed group) of Sm. Parasite growth remained inhibited with prolonged culture in the unwashed group. Interestingly, in the washed group, the growth of parasites was not restored after the removal of Sm, indicating that the protein synthesis mechanism was damaged by 72 h of exposure of Sm. These results also demonstrated that Sm acts as a parasiticidal agent.

To confirm the inhibitory activity of Sm against *Plasmodium* spp., an in vivo rodent malaria test was performed using *P. yoelii* 17XNL- and *P. berghei* ANKA-infected mice. Although Sm is likely inactive when administered orally to mice [7], the reported LD_50_ of the drug following i.p. administration ranged from 170 to 380 μg/kg per injection in mice and the IC_50_ in some murine tumor models varied from 125 to 500 μg/kg [22]. Accordingly, in this study, mice were treated with i.p. Sm at a dose of 100 or 300 μg/kg per injection to avoid any toxic effects. In the uncomplicated rodent malaria model using *P. yoelii* 17XNL-infected mice and a 7-day Sm regimen, no inhibition was observed at a dose of 100 μg/kg (Figure 4A). Meanwhile, partial inhibition of parasite growth was observed in the presence of 300 μg/kg Sm (*p* < 0.05; Figure 4B), in addition to a significantly lower parasitemia peak (18.85%; 11 dpi) than that observed in the control group (40.13%; 12 dpi), without causing mortality. In another set of experiments, the 4- and 7-day Sm regimens prolonged the survival of *P. berghei* ANKA-infected mice (Figure 5) by 33% (up to 25 dpi). The difference between the in vitro and in vivo antiplasmodial activity of Sm possibly resulted from its toxicity and rapid metabolic clearance [37,38], which does not allow adequate drug levels to be reached in vivo. A difference between in vitro and in vivo results was also reported for the antitumor activity of Sm [22].

The use of Sm was discontinued from a phase I clinical study because of a finding that, by daily doses of Sm, two out of five patients (with advanced carcinoma) developed ring scotoma after 13 days (total dose of 0.24 mg/kg) and 15 days (total dose of 0.15 mg/kg); this effect was later defined as Sm-related retinopathy [12]. Later identification by Ottenheijm et al. [7] reported no observable pathological changes in the retinas of any animal treated with toxic doses of Sm and further suggested that Sm-related retinopathy was caused by a poor general condition of the patients and inappropriate drug schedules. By these facts, the chance to use Sm as an alternative drug in malaria is still open. Considering the in vitro and in vivo effects in this study, further development of an Sm scaffold warrants extensive study to address the cytotoxic effects and maintain, or even increase, the antiplasmodial effectivity.

In summary, Sm exhibited comparably potent antiplasmodial activity as chloroquine and artemisinin in vitro. However, this study is limited to the inhibitory activity of Sm against the blood stage of *Plasmodium* spp.; accordingly, investigation in different parasitic stages may give further information/hints to use this compound as an antiplasmodial agent effectively. Furthermore, a more extensive investigation related to the inhibition of three translationally active subcellular compartments of *Plasmodium* spp. may identify which subcellular compartment was specifically inhibited by Sm. Furthermore, the cytotoxic and partial inhibition of parasite growth in vivo of Sm highlights the need for structure-activity studies of Sm derivatives against malaria infection to achieve a better therapeutic index than the parental compound.

## Figures and Tables

**Figure 1 pharmaceutics-14-00544-f001:**
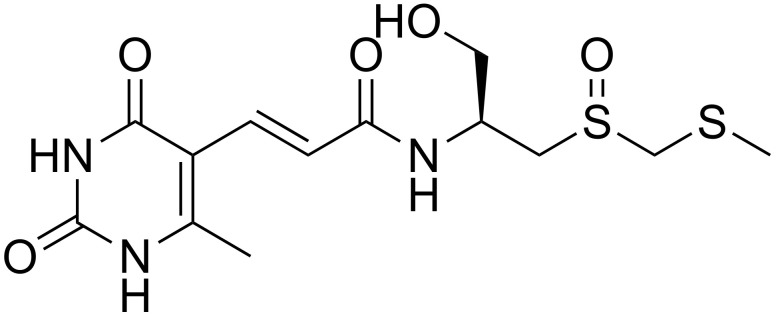
The structure of sparsomycin.

**Figure 2 pharmaceutics-14-00544-f002:**
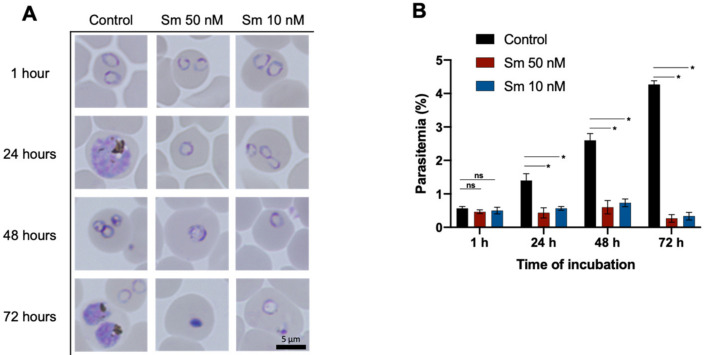
Representative morphology and parasitemia levels of parasites under no treatment (control) or sparsomycin (Sm) treatment (50 nM and 10 nM). (**A**) Parasite morphology after 1, 24, 48 and 72 h of incubation. The parasites in the treated group were retained in the ring stage, they were shrunken and they could not form trophozoites in the life cycle. Scale bar: 5 μm. (**B**) Parasitemia levels after 1, 24, 48 and 72 h of incubation. Parasitemia levels are presented as the mean of triplicate wells and the error bars represent standard deviations. *, significant (*p* < 0.05); ns, not significant (*p* > 0.05). The significance of differences in the level of parasitemia of Sm-treated cultures compared with control were analyzed by two-way ANOVA and a Tukey’s multiple comparison test.

**Figure 3 pharmaceutics-14-00544-f003:**
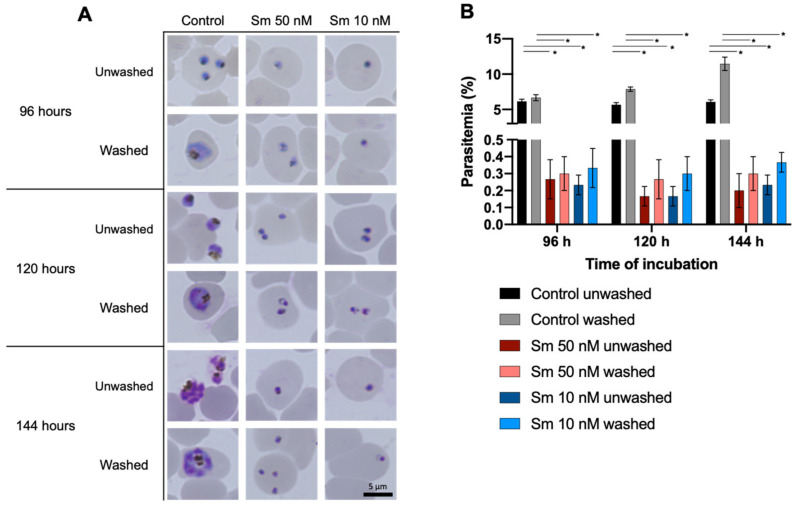
Representative morphologies and parasitemia levels of parasites after 72 h exposure of growth under no treatment (control) or sparsomycin (Sm) treatment (50 and 10 nM). (**A**) Parasite morphologies after 96, 120 and 144 h incubation in the presence (unwashed) or absence (washed) of Sm. All parasites in the treated group were retained in the ring stage, they were shrunken and they could not form trophozoites in the life cycle. Scale bar: 5 μm. (**B**) Parasitemia levels after 96, 120 and 144 h of incubation in the presence (unwashed) or absence of Sm (washed). Parasitemia levels are presented as the mean of triplicate experiments and the error bars represent standard deviations. The significance of differences in the level of parasitemia of Sm-treated cultures compared with control were analyzed by two-way ANOVA and a Tukey’s multiple comparison test (*, significant at *p* < 0.05).

**Figure 4 pharmaceutics-14-00544-f004:**
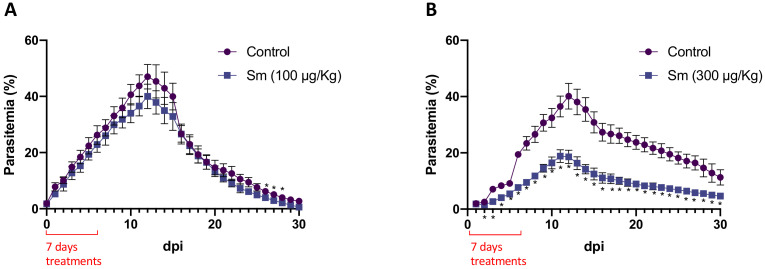
Effects of 7 days of sparsomycin (Sm) treatment on C57BL/6 mice infected with *P. yoelii* 17XNL. (**A**) Parasitemia levels after treatment with 100 μg/kg Sm and (**B**) parasitemia levels after treatment with 300 μg/kg following the inoculation of 1 × 10^7^ infected erythrocytes. Each group consisted of six mice. *: the significance of differences in the level of parasitemia in Sm-treated mice compared with control mice was analyzed by two-way ANOVA followed by Sidak’s multiple comparison test (*p* < 0.05).

**Figure 5 pharmaceutics-14-00544-f005:**
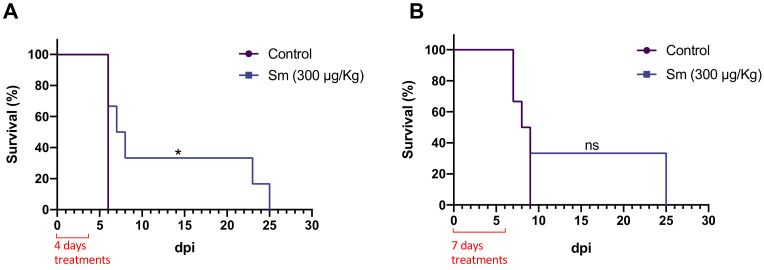
Effects of (**A**) 4 or (**B**) 7 days of treatment with 300 μg/kg sparsomycin (Sm) on the survival of C57BL/6 mice infected with *P. berghei* ANKA. Mice were infected via inoculation with 1 × 10^7^ infected erythrocytes. Each group consisted of six mice. *, significant (*p* < 0.05); ns, not significant (*p* > 0.05). Survival rates were calculated using the log rank (Mantel–Cox) test.

**Table 1 pharmaceutics-14-00544-t001:** Summary of the in vitro activities of sparsomycin compared with the positive controls *.

Compound	IC_50_ *P. falciparum* (nM)		Resistance Index	CC_50_ HFF (μM)	SI for *P. falciparum*		RBC Hemolysis Rate (%) at 100 μM
	3D7	K1			3D7	K1	
Sparsomycin	12.07 ± 4.41 *^a^*	25.43 ± 8.15 *^a^*	2.1	1.14 ± 0.03	94.45	44.83	1.04 ± 0.23
Artemisinin	13.18 ± 2.66 *^a^*	19.89 ± 1.51 *^a^*	1.4	153.00 ± 30.76	10,983.49	7692.31	1.03 ± 0.46
Chloroquine	26.20 ± 3.66 *^b^*	740.07 ± 95.67 *^b^*	28.24	20.71 ± 6.80	790.46	27.98	0.71 ± 0.35

* Values are presented as the mean ± SD of three independent experiments. IC_50_, half-maximal inhibitory concentration. CC_50_, half-maximal cytotoxic concentration. Resistance index, ratio between the IC_50_ values of *P. falciparum* K1 and 3D7. SI, selectivity index (ratio between IC_50_ and CC_50_). HFF, human foreskin fibroblast. RBC, red blood cell. The different letters (*a* or *b*) in the IC_50_ values indicate statistically significant differences among test compounds (one-way ANOVA followed by Tukey’s multiple comparison test; *p* < 0.05).

## Data Availability

Not applicable.

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
