# Peer review of "Sparsomycin Exhibits Potent Antiplasmodial Activity In Vitro and In Vivo"

_pharmaceutics, 2022, doi:10.3390/pharmaceutics14030544_

Round 1

Reviewer 1 Report

The manuscript deals with an interesting topic focused on the antiplasmodial evaluation of sparsomycin. Some minor revisions are nevertheless required please.

General comments:

  • Words in foreign language such as via, in vitro, in vivo, versus should appear in italics throughout the text please.
  • The abbreviation for “milliliter” is mL in the international nomenclature. Please change throughout the text.

Introduction:

  • Line 33: Plasmodium is a genus, not a family.
  • Line 34: An hyphen is required within a word composed, between two vowels (socio-economically).
  • Line 55: Please give some bibliographic references related to "the side effects and cytotoxicity of sparsomycin".

Materials and Methods:

  • Line 72: Please detail the multidrug resistance of the P. falciparum K1 strain: Is it resistant to artemisinin in particular? Indeed, it could have an influence on the interpretation of your results. 
  • Line 94: Please change "d-sorbitol" by D-sorbitol.
  • Lines 138 and 140: "100 µL (...) [were] added/transferred" (not "was").
  • Line 152: "0 [day]" (not days).

Results: Line 165: CC50 with 50 in index.

Discussion:

  • Line 245: Please change "superior" by better. 
  • Line 281: kg and not "Kg".
  • Line 285: "4- and 7-[days]" (and not "day")

References:

  • Reference 24: "et al." should be avoided. If you maintain it, please write it in italics.
  • References 24 and 26: The publication year should appear in bold.
  • References 1, 24 and 26: Why is the DOI not indicated?

Author Response

Reviewer 1

The manuscript deals with an interesting topic focused on the antiplasmodial evaluation of sparsomycin. Some minor revisions are nevertheless required please.

Response: Thank you for your comments concerning our manuscript (Ms. ID:  pharmaceutics-1601418). We would like to thank you for careful and thorough reading our manuscript and for the thoughtful comments and constructive suggestions, which help to improve the quality of our manuscript. We have studied all comments carefully and have revised our manuscript which we hope meet your approval.

General comments:

  • Words in foreign language such as viain vitroin vivo, versus should appear in italics throughout the text please.

Response: The words style was revised to appear in italics throughout the MS.

  • The abbreviation for “milliliter” is mL in the international nomenclature. Please change throughout the text.

Response: The abbreviations of ml were changed to mL throughout the MS.

Introduction:

  • Line 33: Plasmodium is a genus, not a family.

Response: The word family was replaced with genus.

  • Line 34: An hyphen is required within a word composed, between two vowels (socio-economically).

Response: A hyphen was added to the word socio-economically.

  • Line 55: Please give some bibliographic references related to "the side effects and cytotoxicity of sparsomycin".

Response: The references were added. Ref. 6 and 12.

Materials and Methods:

  • Line 72: Please detail the multidrug resistance of the P. falciparum K1 strain: Is it resistant to artemisinin in particular? Indeed, it could have an influence on the interpretation of your results.

Response: P. falciparum K1 is resistant to chloroquine, sulphadoxine, pyrimethamine, and cycloguanil, but not to artemisinin. By using chloroquine and artemisinin, as positive control, we confirmed the effectiveness of our assay system both for P. falciparum 3D7 and K1. The detail of the multi-drug resistance of this strain was added.

  • Line 94: Please change "d-sorbitol" by D-sorbitol.

Response: The letter d- was replaced with D-.

  • Lines 138 and 140: "100 µL (...) [were] added/transferred" (not "was").

Response: The words “was” were replaced with “were”.

  • Line 152: "0 [day]" (not days).

Response: The letter “s” was deleted.

Results: Line 165: CC50 with 50 in index.

Response: The style was changed.

Discussion:

  • Line 245: Please change "superior" by better.

Response: The word superior was replaced with better.

  • Line 281: kg and not "Kg".

Response: The abbreviation was corrected.

  • Line 285: "4- and 7-[days]" (and not "day")

Response: The words were corrected.

References:

  • Reference 24: "et al." should be avoided. If you maintain it, please write it in italics.

Response: The reference style was edited to shows all authors. Changed as ref. 32.

  • References 24 and 26: The publication year should appear in bold.

Response: The style for publication year was corrected. Changed as ref. 32 and 34.

  • References 1, 24 and 26: Why is the DOI not indicated?

Response: The DOI for references 1, 24, and 26 were added. Changed as ref 1, 32, and 34.

Reviewer 2 Report

This study evaluated the growth-inhibitory effects of sparsomycin against Plasmodium falciparum 3D7, P. falci- 15 parum K1, P. yoelii 17XNL, and P. berghei ANKA. There are some points in this manuscript that should be addressed as follows:

  • Introduction:
  1. The reason for selecting sparsomycin among other inhibitors of peptide bond formation for this study should be clarified.
  2. The possible adverse effects of sparsomycin should be addressed.
  3. The possible mechanisms of drug resistance of malaria parasites should be mentioned.
  4. Paragraph 3: To which ribosomal subunit does sparsomycin bind to (60S, 50S, etc…..)?
  5. The reason for selecting human 57 malaria parasite P. falciparum and the rodent malaria parasites P. yoelii and P. berghei to test the effect of sparsomycin should be mentioned.
  • Materials and methods:
  1. CAS and catalogue numbers of the used drugs, kits and chemicals should be mentioned.
  2. The method of quantification of parasitemia and morphology of P. falciparum 3D7 treated with Sm in vitro should be mentioned.
  3. A reference for the equation used for assessment of “The erythrocyte hemolysis rate” should be added.
  4. In the in vivo study, how did you know that the parasitemia level reached 1%?
  5. Data about the housing conditions of mice (Temperature, humidity, etc ….) should be mentioned.
  6. The methods of statistical analysis should be mentioned at the end of “Materials and methods” section.
  7. I think that the number of animals used in this study is too small to reach valid significant results.
  8. References for the used doses of drugs and duration should be added.
  • Results:
  1. Marks of the level of significance should be inserted in table 1, figure 2 and figure 3.
  2. Scale bars should be added to figures 2 and 3.
  • Discussion:

More details should be added to explain the results of the present study.

  • Conclusion:

- I think that the conclusion is not sufficient. The possible clinical implications of the results of the present study should be addressed. Also, limitations of the present study should be mentioned.

  • General comments:
  1. The manuscript should be checked regarding the grammatical and typing errors and plagiarism.
  2. The meaning of the abbreviations should be clearly addressed at their first mention.

Author Response

Reviewer 2

This study evaluated the growth-inhibitory effects of sparsomycin against Plasmodium falciparum 3D7, P. falciparum K1, P. yoelii 17XNL, and P. berghei ANKA. There are some points in this manuscript that should be addressed as follows:

Response: Thank you for your comments concerning our manuscript (Ms. ID:  pharmaceutics-1601418). We would like to thank you for careful and thorough reading our manuscript and for the thoughtful comments and constructive suggestions, which help to improve the quality of our manuscript. We have studied all comments carefully and have revised our manuscript which we hope meet your approval.

  • Introduction:
  1. The reason for selecting sparsomycin among other inhibitors of peptide bond formation for this study should be clarified.

Response: The reason for selecting sparsomycin was added in introduction. Sparsomycin was a hit compound selected by screening of a compound library. Sm exhibited an efficacy at the nanomolar scale, appropriate with our selection criteria (line 49–52).

  1. The possible adverse effects of sparsomycin should be addressed.

Response: The possible adverse effects of sparsomycin were added.

“Despite acute toxicity of Sm in mice [6] and Sm-related retinopathy in the phase I clinical study [12]”… (line 61–62)

  1. The possible mechanisms of drug resistance of malaria parasites should be mentioned.

Response: The possible mechanisms of drug resistance were added in the introduction.

“The spread of multidrug-resistant malaria parasites is broadly caused by four factors, i.e., mutation and/or amplification of the target(s), stress response-based survival mechanisms, removal or sequestration of drugs, and detoxifications [5].” (Line 44–48)

  1. Paragraph 3: To which ribosomal subunit does sparsomycin bind to (60S, 50S, etc…..)?

Response: Sparsomycin was universally binds to 50S and 60S ribosomal subunit. (Line 57)

  1. The reason for selecting human malaria parasite P. falciparum and the rodent malaria parasites P. yoelii and P. berghei to test the effect of sparsomycin should be mentioned.

Response: The reasons for selecting the parasites were added.

“Considering the effects of Sm on protein translation, the in vitro inhibitory activity of Sm against the most common and deadliest human malaria parasite, P. falciparum (3D7 and K1) [15], was evaluated. P. falciparum 3D7 represents a chloroquine sensitive strain, and P. falciparum K1 represents a multi-drug resistant strain, including chloroquine, sulphadoxine, pyrimethamine, and cycloguanil [16]. Furthermore, to confirm the in vitro results, the effects of Sm were also evaluated in vivo using two rodent malaria parasites with different virulence characteristics, P. yoelii 17XNL and P. berghei ANKA, respectively, for uncomplicated/non-lethal and complicated/lethal strains [17].” (Line 63–71)

  • Materials and methods:
  1. CAS and catalogue numbers of the used drugs, kits and chemicals should be mentioned.

Response: CAS and catalogue numbers were added.

  1. The method of quantification of parasitemia and morphology of P. falciparum 3D7 treated with Sm in vitro should be mentioned.

Response: The method was added.

“The parasitemia levels were determined by enumerating the number of infected erythrocytes in relation to uninfected erythrocytes (a minimum of 500 cells were counted).” (Line 145–147)

  1. A reference for the equation used for assessment of “The erythrocyte hemolysis rate” should be added.

Reference: The reference was added; ref. 21.

  1. In the in vivo study, how did you know that the parasitemia level reached 1%?

Reference: We clarified the procedures.

“Approximately 2 h after the injection, 2 μL of blood were collected from the tip of the mice's tail, and the parasitemia levels were checked using Giemsa-stained thin blood smears. When the parasitemia level reached 1%,…”. (Line 194–197)

  1. Data about the housing conditions of mice (Temperature, humidity, etc ….) should be mentioned.

Response: The housing condition were added.

“The mice were allowed to free access to water and food (CLEA Rodent Diet CE-2; Japan CLEA, Tokyo, Japan). The room temperature (24 °C), relative humidity (50%), and lighting (light from 8 AM to 8 PM) were adjusted and controlled.” (Line 188–191)

  1. The methods of statistical analysis should be mentioned at the end of “Materials and methods” section.

Response: The statistical analysis section was added in “Materials and methods”. (Line 205–213)

  1. I think that the number of animals used in this study is too small to reach valid significant results.

Response: The number of animals used in the antiplasmodial test has generally followed the method introduced by Peters et al. (ref. 24, also in ref 23), which were used 5-6 mice per treated group, we follow this common guide line to do in vivo test. To confirm the effects of Sm, we conducted in vivo experiments using different doses or different treatment periods. The data was statistically analyzed. The number of mice used in this study also approved by Institutional Ethics Committee.

  1. References for the used doses of drugs and duration should be added.

Response: The references for the used doses of drug (ref. 22) and duration (ref. 23 and 24) were added.

  • Results:
  1. Marks of the level of significance should be inserted in table 1, figure 2 and figure 3.

Response: The level of significances marks was added.

  1. Scale bars should be added to figures 2 and 3.

Response: The scale bars were added to figures 2 and 3 as well as note in the figure legends.

  • Discussion:

More details should be added to explain the results of the present study.

Response: More detail was added to explain the results of the study.

“The use of Sm was discontinued from phase I clinical study because of a finding that by daily doses of Sm, two out of five patients (with advanced carcinoma) developed ring scotoma after 13 days (total dose 0.24 mg/kg) and 15 days (total dose 0.15 mg/kg), this effect was later defined as Sm-related retinopathy [12]. Later identification by Ottenheijm et al. [7] reported no observable pathological changes in the retinas of any animal treated with toxic doses of Sm and further suggested that the Sm-related retinopathy was caused by a poor general condition of the patients and inappropriate drug schedules. By these facts, the chance to use Sm as an alternative drug in malaria is still open. Considering the in vitro and in vivo effect in this study, further development of Sm scaffold warrants extensive study to address the cytotoxic effects and maintain, or even increase, the antiplasmodial effectivity.” (Line 391–403)

  • Conclusion:

- I think that the conclusion is not sufficient. The possible clinical implications of the results of the present study should be addressed. Also, limitations of the present study should be mentioned.

Response: The possible clinical implications and the limitation of the present study were added in the discussion.

“In summary, Sm exhibited comparably potent antiplasmodial activity as chloroquine and artemisinin in vitro. However, this study is limited to the inhibitory activity of Sm against the blood stage of Plasmodium spp.; accordingly, investigation in different parasitic stages may give further information/hints to use this compound as an antiplasmodial agent effectively. Furthermore, a more extensive investigation related to the inhibition of three translationally active subcellular compartments of Plasmodium spp. may identify which subcellular compartment was specifically inhibited by Sm. Furthermore, the cytotoxic and partial inhibition of parasite growth in vivo of Sm highlights the need for structure-activity studies of Sm derivatives against malaria infection to achieve a better therapeutic index than the parental compound.” (Line 404–413)

  • General comments:
  1. The manuscript should be checked regarding the grammatical and typing errors and plagiarism.

Response: The manuscript was checked regarding the grammatical, typing errors, and plagiarism.

  1. The meaning of the abbreviations should be clearly addressed at their first mention.

Response: The meanings of the abbreviations were re-checked and addressed at the first mention.

Round 2

Reviewer 2 Report

The authors had addressed all my comments.